# The effect of different irrigation activation techniques on smear layer removal, bioceramic sealer penetration, and interfacial adaptation: SEM and CLSM evaluation

Maryam Saber Mahdi◉⊙*, Ranjdar Mahmood Talabani◉⊙

Conservative Department, College of Dentistry, University of Sulaimani, Sulaimani, Kurdistan Region, Iraq

⊙ These authors contributed equally to this article.
* maryam.mahdi@univsul.edu.iq

## Abstract

### Background

The success of endodontic treatment relies on the cleaning, disinfection, and sealing of the root canal system. This study evaluated the effects of four different irrigation techniques on smear layer removal, dentinal tubule penetration, and interfacial adaptation of a bioceramic root canal sealer.

### Method

Eighty-four sound single-rooted lower premolars were decoronated, prepared to medium size WaveOne Gold reciprocating files, and divided randomly into four groups: group I conventional needle irrigation, group II heat activation, group III Diode laser activation, and group IV XP-endo Finisher file activation. Half of the samples (n = 40) were used to assess smear layer removal, while the other forty-four teeth were used to measure the sealer's penetration and interfacial adaptation after obturation with Bio_C sealer and gutta-percha; the assessments were done using scanning electron microscopy (SEM) and confocal laser scanning microscope (CLSM).

### Results

No statistically significant differences were found between irrigation activation groups related to smear layer removal assessment (p-value>0.05), while the XP-endo Finisher file activation group showed significantly the highest sealer penetration in the apical third, and the best adaptation in the middle third.

### Conclusion

Irrigation activation can enhance the penetration of the sealer and interfacial adaptation.

**Data availability statement:** All relevant data are within the manuscript and its Supporting Information files.

**Funding:** The author(s) received no specific funding for this work.

**Competing interests:** No authors have competing interests.

## Introduction

Endodontic treatment is critical for preserving natural teeth with compromised pulp; the root canal system must be properly cleaned, shaped, and sealed to stop reinfection and encourage healing [1].

Proper irrigation is essential in endodontic treatment, as it facilitates the removal of the smear layer from the root canal system, enhances disinfection, prepares the canal for obturation, improves the sealer's penetration and adaptation, and increases the overall success rate of endodontic treatments [2].

On the canal walls, a smear layer composed of organic and inorganic material develops during mechanical instrumentation. This layer includes bacteria, necrotic tissue, and their byproducts, which might hinder the adhesion of obturation materials to the canal walls and prevent disinfectants and intracanal medications from penetrating the dentinal tubules. As a result, its removal is necessary for an endodontic procedure to be successful [3].

A number of techniques and irrigant delivery systems have been developed to improve canal cleanliness following mechanical instrumentation and to increase the efficacy of chemical disinfection inside the root canal system [4]. These include laser activation (Er:YAG, diode laser, etc.), heat, special endodontic files (XP-endo Finisher and self-adjusting files), manual dynamic activation (MDA), passive ultrasonic activation (PUA), and other methods [5,6].

Conventional needle irrigation is the most commonly used irrigation technique. It involves a standard syringe and an endodontic irrigation needle. The needle is inserted into the canal, and the irrigant is delivered using apical-coronal movements. While this method is effective, it may not adequately remove the smear layer from the canal walls, especially in curved canals, and can leave debris in areas that are difficult to reach [7].

In contrast, heat activation uses a specialized handpiece that heats the irrigation solution, typically sodium hypochlorite, to enhance its efficacy. Heating can improve the irrigant's fluid dynamics, promoting better penetration into dentinal tubules that can enhance smear layer removal [8].

Laser activation is a technique that utilizes a diode laser or other type of laser to energize the irrigant within the canal. The laser light interacts with sodium hypochlorite, creating cavitation and acoustic streaming, enhancing the removal of debris and smear layer. Laser activation can be beneficial in achieving more thorough disinfection and cleaning, particularly in complex canal systems [9].

The XP-endo Finisher file activation involves a rotary file designed specifically for enhancing irrigation activation, which creates mechanical agitation that helps displace debris and facilitate better sealer penetration when used with sodium hypochlorite. The file's flexible design allows it to adapt to the canal's shape, making it effective for cleaning even in curved canals. This method has been shown to significantly improve the removal of the smear layer and enhance the interfacial adaptation of sealers [10].

Following proper cleaning and disinfection, root canals are obturated using gutta-percha in combination with endodontic sealers. These sealers fill voids,

irregularities, and accessory canals while enhancing the adhesion between gutta-percha and the dentinal walls. Among available options, bioceramic sealers have gained attention due to their favorable physicochemical properties, including excellent biocompatibility, antimicrobial activity, small particle size, and superior sealing ability [11,12].

Sealer penetration into the dentinal tubules plays a critical role in the long-term success of endodontic therapy. This process is influenced by several factors, such as the presence or absence of a smear layer, dentin permeability, sealer viscosity, film thickness, flow characteristics, setting time, obturation technique, surface tension, capillary action, and solubility [13]. While Grossman emphasized that the ease of sealer removal is a desirable feature of root canal materials [14], an ideal sealer must also achieve reliable adhesion to dentin either through mechanical interlocking, chemical bonding, or both, to prevent bacterial leakage and resist functional stresses during mastication and post-endodontic procedures [15].

This study aims to compare smear layer removal, bioceramic sealer penetration, and interfacial adaptation using four irrigation techniques—conventional needle irrigation, heat activation, diode laser activation, and XP-endo Finisher file activation—evaluated by scanning electron microscopy (SEM) and confocal laser scanning microscopy (CLSM). The null hypothesis states that there are no differences between different irrigation techniques in terms of smear layer removal, bioceramic sealer penetration, or interfacial adaptation.

## Methodology

### Selection of the samples

Extracted human teeth used in this study were obtained from anonymous individuals who had their teeth extracted for orthodontic purposes. All samples were fully anonymized before use, and no personal identifying information was collected or available to the researchers. The study protocol was reviewed and approved by the Ethics Committee of the College of Dentistry, University of Sulaimani, under the code number COD-EC-24–0028, on 16/12/2024, and the requirement for informed consent was waived due to the use of anonymized, discarded biological material.

The sample size of the current study was calculated using G*Power 3.1.9.6 software. For the first part of the study (smear layer removal), the power of the sample size was set at $\alpha = 0.05$, with an effect size of 0.55, where a sample of 40 teeth gave a power of 0.84. For the second part (dentinal tubule penetration and interfacial adaptation), the power of the sample size was calculated at $\alpha = 0.05$, with an effect size of 0.6, where a sample of 40 teeth gave power of 0.85, and the minimum acceptable power was 0.80; therefore, the total sample size was 84 teeth for this study. The collected samples were preserved in normal saline until they were accessed for research purposes starting on 26/12/2024.

The collected teeth were disinfected with 2.5% sodium hypochlorite for 30 minutes [16], and then checked by periapical x-ray in buccolingual and mesiodistal directions to confirm the presence of a single straight canal; canal curvature was measured by using the Schneider method by drawing two lines, the first line was parallel to the long axis of the tooth while the second line was drawn from the apical foramen to intersect the first line and then the angle was measured using Image J software, any tooth with curvature >20° was excluded [17]. The inclusion criteria were extracted sound permanent mandibular premolar teeth, mature apex, single-rooted teeth with one canal, fully developed root canal with no calcification, caries-free teeth, and not previously endodontically treated. In contrast, teeth with multiple canals, severe root curvature, open apex, and roots shorter than 11 mm were excluded.

### Root canal preparation

To standardize the working length and facilitate instrumentation, the teeth were marked to 15 mm using a Vernier caliper, stabilized in a bench vice, and then sectioned using a double-sided diamond disc with a low-speed straight handpiece perpendicular to the long axis of the teeth under copious irrigation [18]. Working length was determined using #10 K-file (Dentsply Sirona, Ballaigues, Switzerland) inserted in the canal until it appeared from the apical foramen, then 1 mm was minimized from the K-file length, the apices were covered by baseplate wax, and the roots embedded in the clear plastic

test tubes filled with putty type silicone impression material (Zeta plus, Zhermack, Italy) to mimic clinical condition of the teeth and prevent irrigation seepage.

Root canal instrumentation was done using the WaveOne Gold reciprocation file system (Dentsply Sirona, Ballaigues, Switzerland) until medium size (35\06) connected to an endomotor (X-Smart IQ, Dentsply Sirona, Ballaigues, Switzerland). The WaveOne Gold system is designed to operate at a speed of 350 RPM, with a reciprocating motion of 150° counterclockwise (CCW) and 30° clockwise (CW), completing a full 360° rotation in three cycles [19]. Before inserting any files, the canals were irrigated using 2 ml of 5.25% NaOCl (Aqua Septica, Aqua Medikal, Istanbul, Turkey) as a lubricant, K-file #10, #15, primary file and medium file of WaveOne Gold system, between each file and another 2 ml of 5.25% NaOCl was used. After chemomechanical instrumentation, the teeth were irrigated with 2 ml of normal saline, 5 ml of 17% EDTA (PPH Cerkamed, Stalowa Wola, Poland), 5 ml of normal saline, 5 ml of NaOCl, and finally 5 ml of normal saline to stop the action of NaOCl. All the irrigation was done using a 27-G irrigation needle connected to a root canal irrigator (VRN, China) to control the flow rate (5 ml/1 min.), 1 mm shorter than the apex in up and down movement [18].

### Irrigation techniques

After completion of instrumentation, all teeth were randomly assigned to one of four experimental groups (n = 21): conventional needle irrigation, heat activation, Diode laser activation, and XP-endo Finisher file activation, as follows:

In group I, the irrigation was carried out using 5 ml of 5.25% NaOCl continuously for 1 min, irrigation was achieved with a 27-G conventional irrigation needle (close ended- double side vented) connected to the root canal irrigator, with up-and-down movements, and followed by 5 ml of distilled water [20].

In group II, 5 ml of 5.25% NaOCl was used; the Gutta Percha obturation pen (Fi-P, Woodpecker, Guangxi, China) was set at 180°C, which takes 0.5 seconds to heat up; the tip size of 35\04 was used, 3 mm shorter than the working length, and activated for 8 seconds without touching the dentinal walls; this activation process was repeated 10 times, with 0.5 ml of a fresh solution at each cycle, followed by 5 ml of distilled water [20].

In group III, for laser activation, this study used a Diode laser (Solase, Lazron, USA) with a fiber optic tip (diameter = 200 μm/976 nm); this device was connected to a phone application through Bluetooth, and irrigation activation mode was chosen in the pulse mode with a maximum output of 12 W and average power of 1.2 W for 5 seconds per cycle, introduced till 1 mm short of the apex and recessed in helicoidal motion, repeated four times at intervals of 10 seconds. In each cycle sodium hypochlorite was refreshed with 1.25 ml to the total of 5 ml, followed by 5 ml of distilled water as the final flush [21].

In group IV, activation was performed with 5 ml of 5.25% sodium hypochlorite using the XP-endo Finisher (FKG, Switzerland) connected to the endomotor at 800 rpm speed and 1 Ncm torque for one minute with up-and-down movement while touching the walls, followed by 5 ml of distilled water [18].

All the teeth had the same irrigation volume during the sample preparation for standardization.

### Evaluation of smear layer removal

Forty teeth were chosen randomly (ten from each group) for the evaluation of smear layer removal after activation was finished. The canal orifice was covered with a moistened cotton pellet, and a groove was made with a diamond disc in a buccolingual direction until a thin shell of dentin was preserved. This was followed by the separation of the root with a chisel (to stop the smear layer formed from the cutting from entering the canal); after sectioning, only one section was chosen randomly [22].

After that, these sections were fixed using 2.5% buffered glutaraldehyde (EOBA CHEMIE PVT, India) and 0.1M sodium cacodylate (BDH Chemicals Ltd, England) (pH = 7.4) at 4°C for 12 hours. They were then dehydrated using ascending alcohol and metalized with a coating of gold under a vacuum after being attached to metal stubs [23].

By using SEM (ThermoFisher, Axia, ChemiSEM, USA), three areas were examined 3,6,9 mm away from the apex to represent the thirds (apical, middle, and coronal) under 3000x magnification, using the scoring system proposed by Gambarini and Laszkiewicz, score 1: no smear layer, dentinal tubules open, score 2: small amount of smear layer and some dentinal tubules open, score 3: a homogenous smear layer covers the root canal wall, with only a few dentinal tubules open, score 4: complete root canal wall covered by a homogenous smear layer, no open dentinal tubules, and score 5: heavy, non-homogenous smear layer covering the complete root canal wall [24].

### Sealer's penetration and interfacial adaptation evaluation

The other forty teeth were obturated using Bio-C sealer (Angelus, Londrina, Brazil) with the corresponding medium-size Gutta-percha (Dentsply Sirona, Ballaigues, Switzerland), after drying with three paper points (Dentsply Sirona, Johnson City, USA), and periapical x-ray was taken for each root in buccolingual direction to check the quality of obturation, then these samples were incubated for one week at 37°C to ensure complete setting of the sealer [23].

These samples were then cut 3,6,9 mm away from the apex horizontally, perpendicular to the long axis of the root, to represent the thirds; the same procedure as for fixation, dehydration, and metallization of the smear layer part was used. In each section, the maximum penetration, minimum penetration of the sealer, and the maximum gap between the dentin and the sealer were measured in µm, under 800x magnification for penetration and under 1000x magnification for interfacial adaptation, and the measurements were done using Image J software (Version 1.54 g, Wayne Rusband and Contributors, National Institutes of Health, USA) [5,25].

For the confocal laser scanning microscope (Labomed, California, USA) four samples were randomly selected one from each group, and the sealer was mixed with 0.1% Rhodamine B fluorescent dye (GCC, UK), the sectioning was done in 3,6,9 mm away from the apex in 1 mm thickness, by using pictures of 0.4µm stepsize under 5x magnification [26,27].

### Statistical analysis

The research was analyzed using SPSS software (IBM SPSS Statistics, V 26; IBM Corp, Armonk, NY, USA). The Shapiro-Wilk test results showed non-normally distributed data for all parts of the study (smear layer, bioceramic sealer penetration, and interfacial adaptation) separately, allowing for the selection of the non-parametric tests; Kruskal-Wallis test was done for all parts, and Dunn's post hoc test was used for multiple comparisons when significance was present. For all analyses, a 5% level of statistical significance was used.

## Results

### Reliability tests

Regarding the smear layer removal assessment part, the assessment was done by two examiners blindly. The Weighted Kappa test was done, and the result was 0.82, indicating very good reliability between the two examiners, and the mean of the two readings was taken [28].

Two readings each were done for the maximum and minimum bioceramic sealer levels by the same examiner at two different times; the second reading was done 15 days after the first reading. The intra-examiner reliability test (interclass correlation coefficient) was done, and the result was 0.86, which indicates a good reliability between the two readings. The mean of the two readings was used for data analysis [29].

### Smear layer

The results of the smear layer assessment are shown in Fig 1; the resultant pictures were arranged according to the groups and thirds.

When the Kruskal-Wallis non-parametric test was used to compare the thirds inside each group, the result was significant (P-value<0.05), which means that there was a difference between the thirds. When the multiple comparisons were

| Group | Coronal third | Middle third | Apical third |
|-------|---------------|--------------|--------------|
| I | (a) | (b) | (c) |
| II | (d) | (e) | (f) |
| III | (g) | (h) | (i) |
| IV | (j) | (k) | (l) |

**Fig 1. Representative images of each group and third with the scoring; orange arrows represent open dentinal tubules, while blue arrows represent closed dentinal tubules under 3000x magnification, the scores are: a) score 2, b) score 2, c) score 2, d) score 3, e) score 2, f) score 2, g) score 2, h) score 2, i) score 3, j) score 3, k) score 2, and l) score 3.**

done using post-hoc Dunn's test corrected by Bonferroni, the result was significant differences between apical and coronal thirds. Kruskal-Wallis was used to compare the groups' thirds between different groups, and no significant difference was found between them (p-value>0.05), as shown in Tables 1 and 2.

**Bioceramic sealer penetration**

The resultant pictures of the bioceramic sealer penetration are shown in Fig 2.

According to the Shapiro-Wilk test, the data were not normally distributed; therefore, the Kruskal Wallis test was used. A significant difference was found between the apical thirds of different groups, as shown in Table 3. For further comparisons, Dunn's post hoc was done, and significant differences were found between groups I and III, and I and IV, as shown in Table 4.

**Interfacial adaptation**

The resultant pictures of interfacial adaptation are shown in Fig 3.

When the Kruskal-Wallis non-parametric test was used to compare groups by thirds, significant differences were found in the middle and apical thirds; therefore, Dunn's test was used for multiple comparisons in the middle and apical thirds. Significant differences were found between group I and IV, II and III, and II and IV in the middle third, while significant differences were found between group I and II, I and IV, and II and III in the apical third, as shown in Tables 5–7.

**CLSM results**

The CLSM analysis was conducted to validate the penetration of the sealer into the dentinal tubules. One representative sample from each group was selected for evaluation. Penetration depth was measured in μm, recording both

**Table 1. The smear layer comparison in each group using hoc Dunn's test corrected by Bonferroni.**

| Group | Thirds | P-value |
|---|---|---|
| Group I | Coronal/ Middle | 0.1033 |
| | Coronal/Apical | 0.0132* |
| | Middle/Apical | 0.397 |
| Group II | Coronal/ Middle | 0.753 |
| | Coronal/Apical | 0.005 ** |
| | Middle/Apical | 0.014 * |
| Group III | Coronal/ Middle | 0.515 |
| | Coronal/Apical | 0.049 * |
| | Middle/Apical | 0.189 |
| Group IV | Coronal/ Middle | 0.413 |
| | Coronal/Apical | 0.004 ** |
| | Middle/Apical | 0.042 * |

* means a significant difference at P-value< 0.05, ** means a highly significant at P-value<0.01, *** means very high significant at P-value< 0.001.

**Table 2. The smear layer groups were compared by thirds' mean rank using the Kruskal-Wallis test.**

| Third | Group I | Group II | Group III | Group IV | Chi-square | P-value |
|---|---|---|---|---|---|---|
| Coronal | 23.30 | 22.50 | 16.90 | 19.30 | 2.084 | 0.555 |
| Middle | 26.2 | 18.35 | 19.90 | 17.55 | 3.750 | 0.240 |
| Apical | 22.35 | 22.20 | 21.35 | 18.10 | 0.782 | 0.854 |

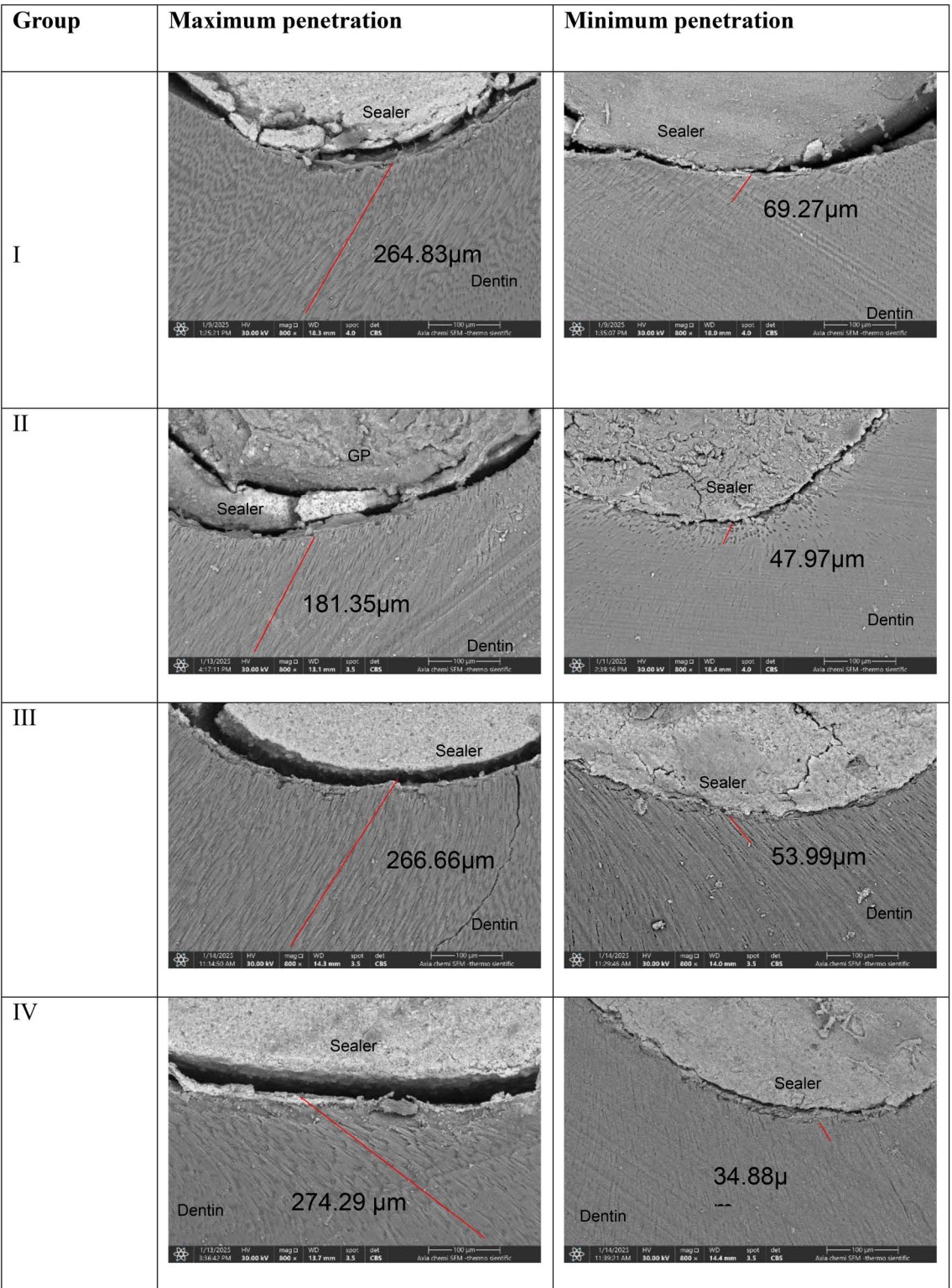

**Fig 2. Representative pictures of bioceramic sealer penetration of maximum and minimum penetration of each group, red lines represent the bioceramic sealer penetration in the dentinal tubules under 800x magnification.**

**Table 3. Group comparison of bioceramic sealer penetration according to thirds using the Kruskal-Wallis test.**

| Third | Group I | Group II | Group III | Group IV | Chi-square | P-value |
|---|---|---|---|---|---|---|
| Coronal | 14.80 | 27.70 | 19.40 | 20.10 | 6.271 | 0.099 |
| Middle | 20.90 | 18.30 | 16.50 | 26.30 | 3.998 | 0.262 |
| Apical | 11.90 | 18.20 | 24.30 | 27.60 | 10.544 | 0.014* |

* means a significant difference at P-value< 0.05

**Table 4. Multiple group comparison of bioceramic sealer penetration in apical thirds by Dunn's post hoc test corrected by Bonferroni.**

| Group | Mean rank difference | P-value |
|---|---|---|
| Group I × Group II | −6.3 | 0.228 |
| Group I × Group III | −12.4 | 0.018 * |
| Group I × Group IV | −15.7 | 0.0027 ** |
| Group II × Group III | −6.1 | 0.243 |
| Group II × Group IV | −9.4 | 0.072 |
| Group III × Group IV | −3.3 | 0.528 |

* means a significant difference at P-value< 0.05, ** means a highly significant difference at P-value<0.01.

maximum and minimum values. Initial observations were made using standard microscopic imaging. Subsequently, a three-dimensional reconstruction was generated by compiling 40 sequential images, each captured at 10 µm intervals, to provide a volumetric view of the sealer distribution, as illustrated in Figs 4 and 5.

## Discussion

Although irrigation is one of the most crucial methods for disinfecting the root canal system and cleaning the preparation and tissue residues that impair the fit between the canal walls and the filling material, approximately 50% of the inner canal walls are not touched by the preparation files. This has an impact on the efficacy of the treatment performed [30].

This study compared three different irrigation activation techniques (heat, Diode laser, and XP-endo Finisher file) with conventional needle irrigation (CNI) to evaluate the removal of the smear layer, penetration of bioceramic sealer into dentinal tubules, and interfacial adaptation of the bioceramic sealer using SEM.

SEM is widely utilized for evaluating smear layer removal due to its ability to generate high-resolution images that provide topographical, morphological, and compositional details of the root canal surface. SEM enables precise visualization of sealer penetration depth, surface integrity, and the interface between sealers and dentinal tubules, making it a valuable tool for assessing the effectiveness of irrigation protocols and obturation materials [31]. Despite these advantages, SEM has limitations. The technique relies on two-dimensional imaging, which may result in misinterpretation due to artifacts. Additionally, sample preparation involves dehydration and exposure to high vacuum, which can lead to structural changes such as material detachment from canal walls or the formation of cracks and distortions in the dentin [32]. Although alternative techniques such as digital image analysis, atomic force microscopy, environmental SEM, and co-site optical microscopy have emerged, each presents its own set of limitations and challenges [33,34].

Overall, the smear layer removal pattern was consistent across irrigation techniques, with differences observed within canal thirds rather than between groups. No significant difference was found between the thirds in the different groups (P-value> 0.05). Still, in the coronal third, the lowest mean rank was observed in the heat activation group, while in the middle and apical thirds, the XP-endo Finisher file had the lowest mean rank, and the conventional needle irrigation

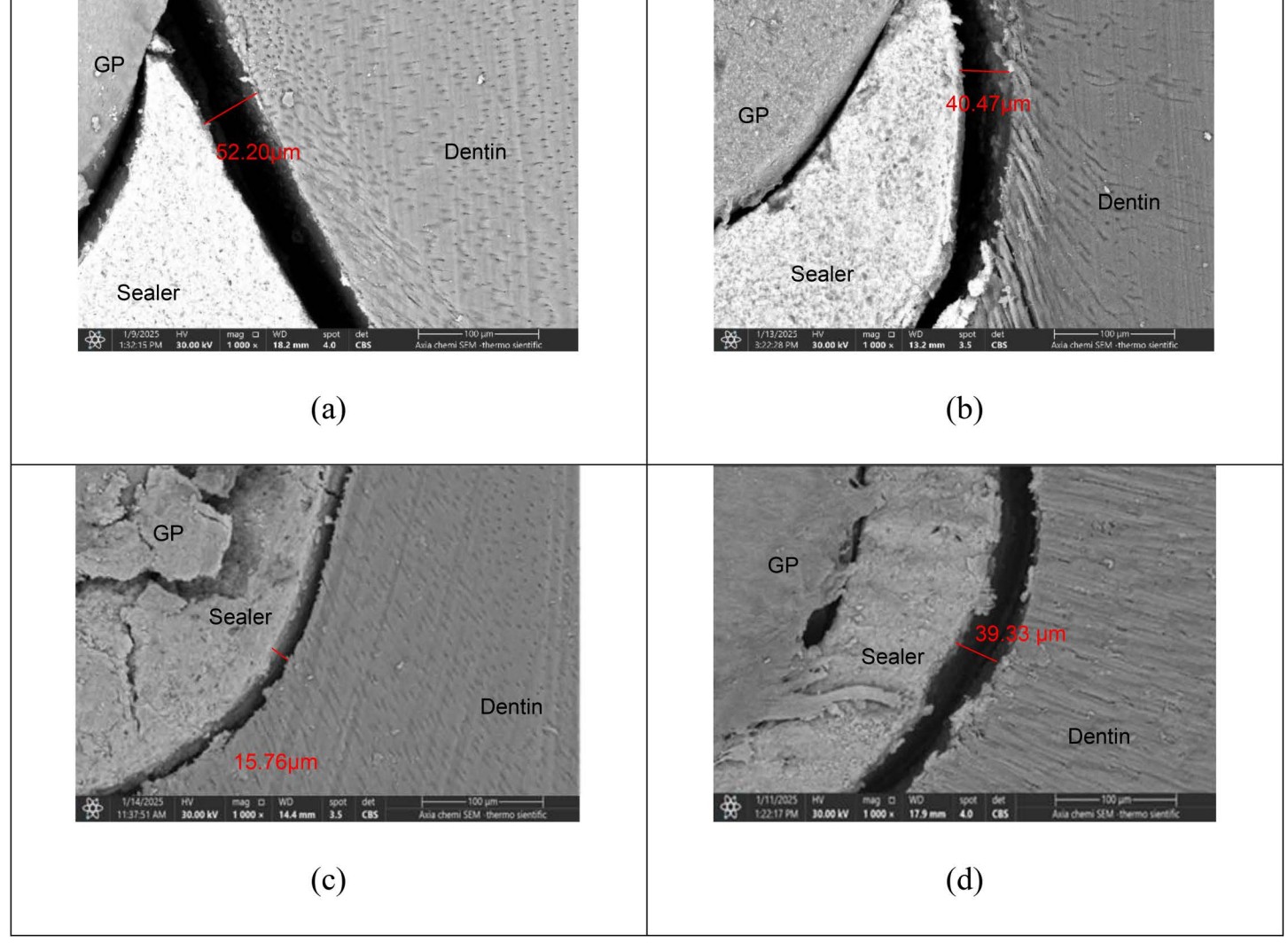

**Fig 3. Representative picture of interfacial adaptation according to groups: a) conventional needle irrigation, b) heat activation, c) Diode laser activation, d) XP-endo Finisher file, the red lines represent the interfacial gap measurement in µm, under 800x magnification.**

**Table 5. Interfacial adaptation groups' comparison using the Kruskal-Wallis test.**

| Third | Group I | Group II | Group III | Group IV | Chi-square | P-value |
|---|---|---|---|---|---|---|
| Coronal | 27.70 | 22.80 | 15.00 | 16.50 | 7.565 | 0.056 |
| Middle | 25.65 | 25.90 | 15.65 | 14.80 | 8.174 | 0.043 * |
| Apical | 11.60 | 29.30 | 14.80 | 26.30 | 16.301 | 0.001 ** |

* means a significant difference at P-value< 0.05, ** means a highly significant at P-value<0.01.

**Table 6. Interfacial adaptation group comparison in the middle third by Dunn's post hoc test corrected by Bonferroni.**

| Group | Mean rank difference | P-value |
|---|---|---|
| Group I × Group II | −0.25 | 0.96 |
| Group I × Group III | 10 | 0.056 |
| Group I × Group IV | 10.85 | 0.038 * |
| Group II × Group III | 10.25 | 0.0499 * |
| Group II × Group IV | 11.1 | 0.033 * |
| Group III × Group IV | 0.85 | 0.871 |

* means a significant difference at P-value <0.05

**Table 7. Interfacial adaptation group comparison in the apical third by Dunn's post hoc test corrected by Bonferroni.**

| Group | Mean rank difference | P-value |
|---|---|---|
| Group I × Group II | −17.3 | 0.0009 *** |
| Group I × Group III | −4.4 | 0.4 |
| Group I × Group IV | −14.3 | 0.006 ** |
| Group II × Group III | 12.9 | 0.014 * |
| Group II × Group IV | 3 | 0.566 |
| Group III × Group IV | −9.9 | 0.059 |

* means a significant difference at P-value< 0.05, ** means a highly significant at P-value<0.01, *** means very high significant at P-value< 0.001.

retained the highest mean rank across all thirds. These results coincide with research by Živković et al. [35] indicating that because of gas-particle entrapment and complicated anatomy, traditional irrigation systems typically fail to deliver adequate performance to the apical third. Mechanical stimulation (XP-endo Finisher file) efficiently reduces the smear layer in the apical third by eliminating air bubbles that impede penetration due to its unique shape. It can clean the canal more effectively by getting to those hard-to-reach areas. This file's small diameter (ISO 25) and ability to change shape while rotating in the canal (Martensitic and Austenitic phase) enable it to effectively remove dentin debris and the smear layer [35,36].

When each group's thirds were compared, there were significant differences between coronal and apical thirds, and higher mean ranks (more smear layer) were found in the apical thirds than in the coronal thirds due to the difficulty of removing the smear layer in the apical thirds, related to the stagnation plane of residual fluid and gas bubbles [36].

In addition, the heat activation group showed a lower mean rank in the middle third than conventional needle irrigation and Diode laser groups.

Clinically, sealer penetration is important. A proper adaptation of the sealer is made possible by the penetration of sealer tags into dentin. Two positive outcomes of the high penetration, adaptability, and adhesive properties are the increased surface area interaction between the sealer and dentin and the antibacterial effect of trapping remaining germs in dentinal tubules [37].

Two readings were obtained from each section for the penetration of the bioceramic sealer: one for the greatest penetration and one for the minimum penetration, then the mean was calculated. The buccolingual direction was found to have a higher sealer penetration than the mesiodistal direction. The root cross-sections of single-rooted teeth showed a butterfly-like appearance due to increased sclerosis along the tubules on the distal and mesial sides of the canal lumen [38].

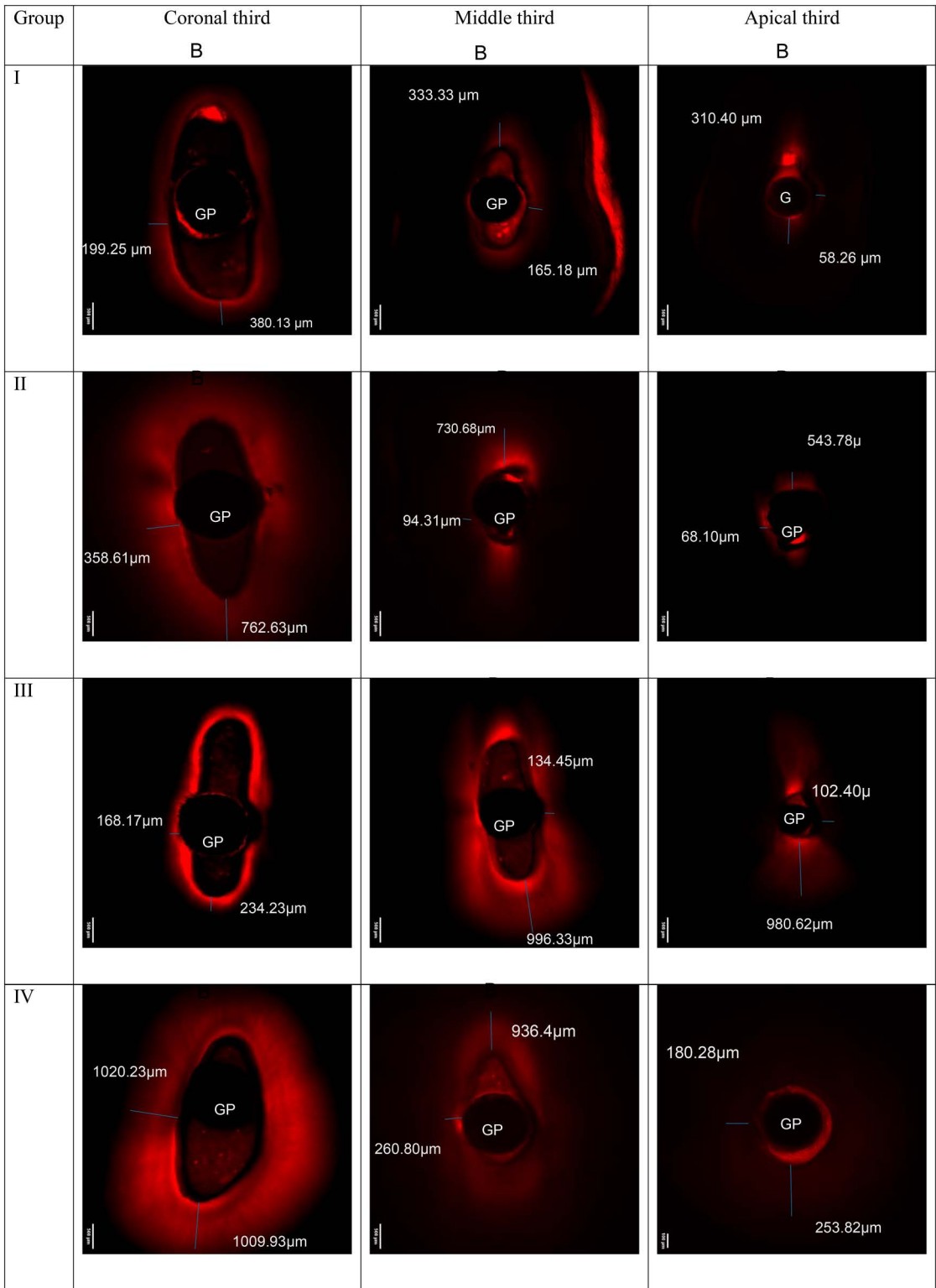

**Fig 4. Representative image obtained using Confocal Laser Scanning Microscopy (CLSM) at 5× magnification.** The field of view is 4 × 4 mm. The excitation wavelength was 561 nm, and the emission was collected at 630–690 nm. The measurements (in µm) indicate the maximum and minimum penetration depths of the bioceramic sealer. B denotes the buccal side of the section, GP denotes Gutta percha.

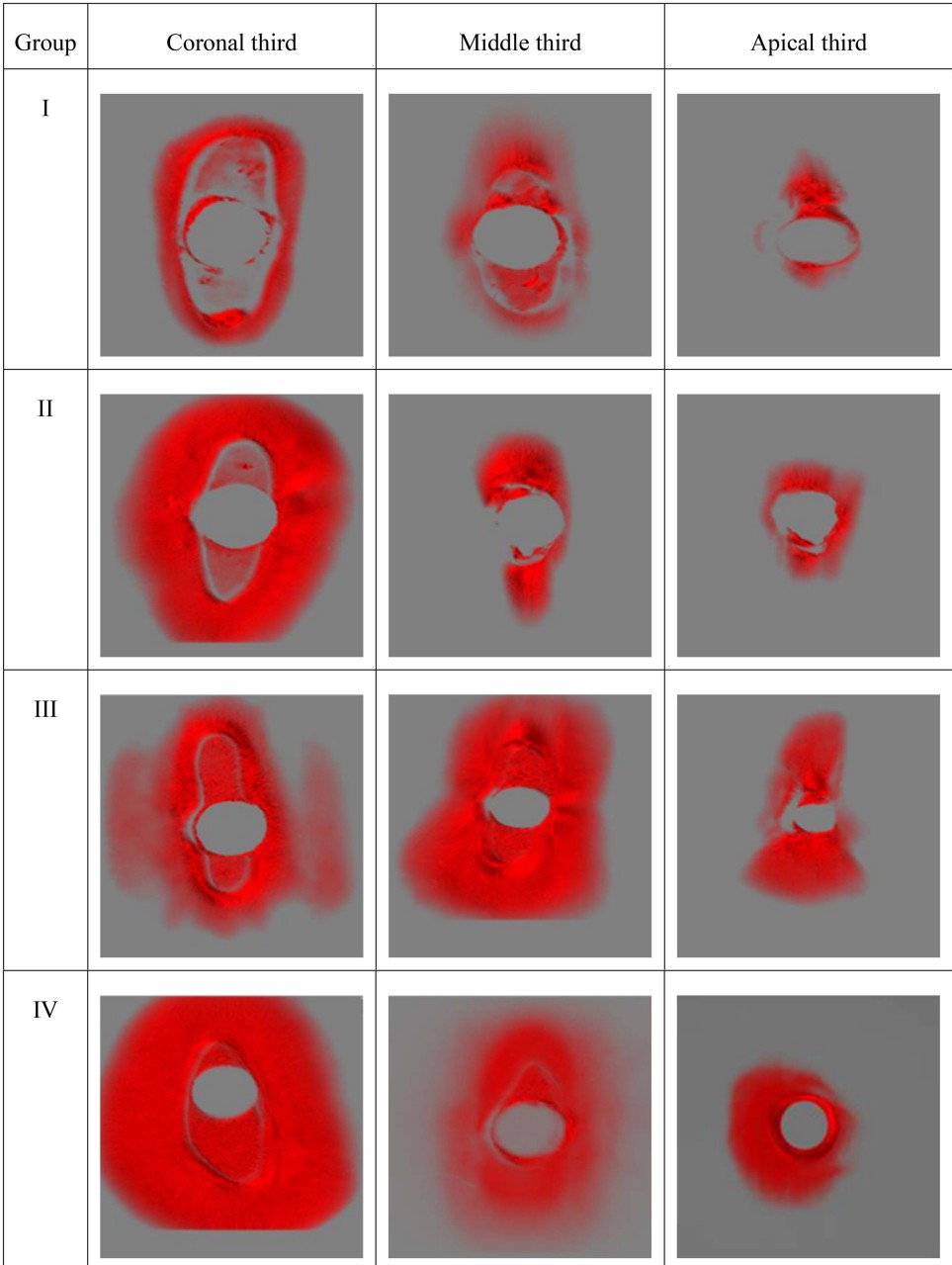

| Group | Coronal third | Middle third | Apical third |
|-------|---------------|--------------|--------------|
| I | | | |
| II | | | |
| III | | | |
| IV | | | |

**Fig 5. Representative pictures of CLSM in volume view.** The red color represents the sealer penetration into dentinal tubules under 5x magnification.

In the current study, in terms of sealer penetration, the highest mean rank was found in the coronal and middle third in comparison to the apical third because of the significant structural variations that can be seen in the apical part of human teeth, including pulp stones, cementum-like lining, areas of resorption, uneven secondary dentin, accessory root canals, and deviations in the apex from the long axis of the root canal. The progressive mineral infiltration of tubule lumens is one of the continual microstructural changes that occur in dentin as people age. Dental sclerosis is the term for this process, which starts at the root apex, moves toward the crown, and reduces the dentin's resistance to fracture. This could help

explain why the current study reported higher smear layer in the apical third and more sealer penetration in the coronal region as opposed to the apical region [39].

Significant differences were found in all groups between the coronal and apical thirds and the middle and apical thirds, except the conventional needle irrigation group; the difference was found only between the middle and apical thirds (coronal and middle thirds had higher penetration).

A significant difference was observed in the apical third among the different groups, with the XP-endo Finisher file showing the highest penetration. Additionally, the XP-endo Finisher group exhibited the highest mean ranks in both the middle and apical thirds, a finding consistent with the results reported by Demir et al. [40]. This may also explain the lower mean ranks for the smear layer observed in the middle and apical thirds of the XP-endo Finisher group. In the coronal third, although the heat activation group demonstrated the highest penetration, the difference was not statistically significant. A study done by Kilic et al. compared sonic activation, passive ultrasonic activation and needle activation in terms of sealer penetration and found that sonic activation showed the highest penetration [41].

The results of the CLSM showed that the CNI group had the lowest penetration among groups, and the apical thirds showed the lowest penetration compared to other thirds, which is also found in the study done by Almasri et al., indicated that irrigation activation enhancd the sealer penetration [26].

Regarding the interfacial adaptation of the bioceramic sealer, the maximum gap (µm) between the sealer and the canal wall was evaluated across different thirds of the root canal. In the CNI group, a significantly lower gap was observed in the apical third compared to the coronal third. Similarly, in the XP-endo Finisher group, the middle third showed a significantly lower gap than the apical third. In contrast, no significant differences were found between the thirds in the heat activation and diode laser groups.

When comparing the same third across groups, XP-endo Finisher showed the lowest gap in the middle third, while CNI showed the lowest gap in the apical third. The diode laser also demonstrated a lower interfacial gap than heat activation in the apical third. No previous research has utilized these activation techniques to assess the interfacial adaptation, and only one study done by Sobhy, has compared CNI and diode laser activation, demonstrated that 980 nm diode laser-activated irrigation significantly improved the adaptability of root canal filling materials in oval-shaped canals, particularly in areas where traditional techniques struggle to achieve complete cleaning and adaptation. This finding aligns with the present results, in which diode laser irrigation showed a reduced interfacial gap in the apical third compared to heat activation, indicating that the laser's ability to improve fluid dynamics and smear layer removal may translate into better sealer penetration and adaptation [42].

The study's limitations related to only a small portion of the canal being examined using SEM for smear layer removal assessment [43], and using SEM for bioceramic sealer penetration and interfacial adaptation is also regarded as a limitation because SEM destroys the sample, dehydrates it, and may cause deformation and sealer loss [44]. CLSM is considered better. Still, the use of Rhodamine B fluorescent dye is also considered a limitation because this dye is hydrophilic, so it gives false positive results. Using Fluor 3 dye is considered better because of the hydrophobicity of the dye [45]. In vitro studies may not provide us with a comprehensive understanding of in vivo conditions. This research, however, may provide a basis for future investigations into the teeth under in vivo conditions employing different irrigation activation methods, a combination of irrigation activation methods, bond strength, and the durability of the sealer over time.

The results of this research have important clinical considerations for endodontic treatment. The best root canal sealing, long-term treatment success, and reinfection prevention depend on the efficient removal of the smear layer, increased penetration of bioceramic sealers into dentinal tubules, and improved interfacial adaptability.

## Conclusions

Within the limitations of the study, non-significant differences were found between irrigation groups in smear layer removal. In contrast, significant differences were found in bioceramic sealer penetration. XP-endo finisher file activation

group showed the highest penetration in the apical third and best adaptation in the middle third compared to other groups, while the conventional needle irrigation showed the best adaptation in the apical third.

## Supporting information

**S1 File.  Raw data.**
(DOCX)

## Author contributions

**Conceptualization:** Maryam Saber Mahdi, Ranjdar Mahmood Talabani.

**Data curation:** Maryam Saber Mahdi.

**Funding acquisition:** Maryam Saber Mahdi.

**Methodology:** Maryam Saber Mahdi, Ranjdar Mahmood Talabani.

**Project administration:** Ranjdar Mahmood Talabani.

**Supervision:** Ranjdar Mahmood Talabani.

**Writing – original draft:** Maryam Saber Mahdi.

**Writing – review & editing:** Ranjdar Mahmood Talabani.

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
