## [Decision Letter · Decision Letter 0]

17 Sep 2025

PONE-D-25-26888The effect of different irrigation activation techniques on smear layer removal, bioceramic sealer penetration, and interfacial adaptation: SEM and CLSM evaluationPLOS ONE

Dear Dr. Mahdi,

Thank you for submitting your manuscript to PLOS ONE. After careful consideration, we feel that it has merit but does not fully meet PLOS ONE’s publication criteria as it currently stands. Therefore, we invite you to submit a revised version of the manuscript that addresses the points raised during the review process.

We look forward to receiving your revised manuscript.

Kind regards,

Mohmed Isaqali Karobari, BDS, MScD.Endo, Ph.D. Endo, FDS, FPFA, FICD, MFDS

Academic Editor

PLOS ONE

Journal Requirements:

2. We note that your Data Availability Statement is currently as follows: 

“All relevant data are within the manuscript and its Supporting Information files.”

**Additional Editor Comments:**

Dear Authors,

Kindly read all the comments given by the reviewers carefully and address them; make the changes in the revised manuscript accordingly.

Best regards and keep well

Reviewers' comments:

Reviewer's Responses to Questions

**Comments to the Author**

1. Is the manuscript technically sound, and do the data support the conclusions?

Reviewer #1: Yes

Reviewer #2: Yes

2. Has the statistical analysis been performed appropriately and rigorously? 

Reviewer #1: Yes

Reviewer #2: Yes

3. Have the authors made all data underlying the findings in their manuscript fully available?

Reviewer #1: Yes

Reviewer #2: Yes

4. Is the manuscript presented in an intelligible fashion and written in standard English?

Reviewer #1: Yes

Reviewer #2: Yes

5. Review Comments to the Author

Reviewer #1: 1.Was the samples analysed for curvature and what method was used?

2. cite recent 2025 article.

3. check for grammatical errors

4. add discussion from 2025 articles

5.the study is performed adequately folllowing protocol

Reviewer #2: In this study, the author(s) comparised the effect of different irrigation activation techniques on smear layer removal, bioceramic sealer penetration, and interfacial adaptation using SEM and CLSM evaluation techniques. According to the my revieew, the experimental design and academic writing are very consistent and well-done. It is a very comprehensive article that will contribute to the literature.

6. PLOS authors have the option to publish the peer review history of their article (what does this mean?). If published, this will include your full peer review and any attached files.

Reviewer #1: No

Reviewer #2: No

---

## [Author Response · Author response to Decision Letter 1]

17 Sep 2025

Manuscript ID: PONE-D-25-26888

Title: The effect of different irrigation activation techniques on smear layer removal, bioceramic sealer penetration, and interfacial adaptation: SEM and CLSM evaluation

Authors: Maryam Saber Mahdi, Ranjdar Mahmood Talabani

Reviewer #1

Comment 1: Was the samples analysed for curvature and what method was used?

Response: Yes, the root canal curvature of all samples was analyzed using the Schneider method, which has now been clearly described in the Materials and Methods section. We have cited Balani et al., 2015 [17] as a reference for this method.

Comment 2: Cite recent 2025 article.

Response: We have added recent 2025 references to the manuscript and discussion:

• Kilic et al., 2025 [41]

• Sobhy, 2025 [42]

Comment 3: Check for grammatical errors.

Response: The manuscript has been thoroughly revised for grammar, punctuation, and language clarity to ensure readability and compliance with PLOS ONE standards.

Comment 4: Add discussion from 2025 articles.

Response: Discussion has been updated to incorporate findings from the 2025 articles [41,42].

Comment 5: The study is performed adequately following protocol.

Response: We thank the reviewer for the positive evaluation of our methodology.

Reviewer #2

Comment: In this study, the author(s) compared the effect of different irrigation activation techniques on smear layer removal, bioceramic sealer penetration, and interfacial adaptation using SEM and CLSM evaluation techniques. According to my review, the experimental design and academic writing are very consistent and well-done. It is a very comprehensive article that will contribute to the literature.

Response: We thank the reviewer for the positive feedback and acknowledgment of the study’s design, methodology, and contribution to the literature.

---

## [Decision Letter · Decision Letter 1]

23 Sep 2025

The effect of different irrigation activation techniques on smear layer removal, bioceramic sealer penetration, and interfacial adaptation: SEM and CLSM evaluation

PONE-D-25-26888R1

Dear Dr. Mahdi,

We’re pleased to inform you that your manuscript has been judged scientifically suitable for publication and will be formally accepted for publication once it meets all outstanding technical requirements.

Kind regards,

Mohmed Isaqali Karobari, BDS, MScD.Endo, Ph.D. Endo, FDS, FPFA, FICD, MFDS

Academic Editor

PLOS ONE

Additional Editor Comments (optional):

Dear Authors,

The authors have addressed all the comments and suggestions reviewers gave, and the manuscript has dramatically improved. The manuscript can be accepted for publication in its current form. I would like to congratulate the authors and wish them all the very best in their future endeavors.

Best regards and keep well.

Reviewers' comments:

Reviewer's Responses to Questions

**Comments to the Author**

1. If the authors have adequately addressed your comments raised in a previous round of review and you feel that this manuscript is now acceptable for publication, you may indicate that here to bypass the “Comments to the Author” section, enter your conflict of interest statement in the “Confidential to Editor” section, and submit your "Accept" recommendation.

Reviewer #1: All comments have been addressed

Reviewer #2: All comments have been addressed

2. Is the manuscript technically sound, and do the data support the conclusions?

Reviewer #1: Yes

Reviewer #2: Yes

3. Has the statistical analysis been performed appropriately and rigorously? 

Reviewer #1: Yes

Reviewer #2: Yes

4. Have the authors made all data underlying the findings in their manuscript fully available?

Reviewer #1: Yes

Reviewer #2: Yes

5. Is the manuscript presented in an intelligible fashion and written in standard English?

Reviewer #1: Yes

Reviewer #2: Yes

6. Review Comments to the Author

Reviewer #1: the manuscript is well written and the concepts are explained well. the newers trends mentioned and prospectve scope mentioned

Reviewer #2: (No Response)

7. PLOS authors have the option to publish the peer review history of their article (what does this mean?). If published, this will include your full peer review and any attached files.

Reviewer #1: No

Reviewer #2: **Yes: **Kürşat Er

---

## [Editor Report · Acceptance letter]

PONE-D-25-26888R1

PLOS ONE

Dear Dr. Mahdi,

I'm pleased to inform you that your manuscript has been deemed suitable for publication in PLOS ONE. Congratulations! Your manuscript is now being handed over to our production team.

Kind regards,

on behalf of

Prof Dr. Mohmed Isaqali Karobari

Academic Editor

PLOS ONE